# Metabolome and transcriptomics analyses reveal quality differences between *Camellia tachangensis* F. C. Zhang and *C. sinensis* (L.) O. Kunzte

Yunfei Xu[1,2☯], Qihang Zhou[3☯], Xinglin Wang[4☯], Xingpan Meng[4], Zhengdong Zhang[5], Xu Zhang[6], Ximin Zhang[4], Suzhen Niu[7], Guiping Chen[8]*, Lunxian Liu[4]*, Tie Shen[1,2,3,4]*

1 Guizhou Key Laboratory of Advanced Computing, Guizhou Normal University, Guiyang, China, 2 School of Cyber Science and Technology, Guizhou Normal University, Guiyang, China, 3 School of Mathematical Sciences, Guizhou Normal University, Guiyang, China, 4 Key Laboratory of National Forestry and Grassland Administration on Biodiversity Conservation in Karst Mountainous Areas of Southwestern China, Engineering Research Center of Carbon Neutrality in Karst Areas, Ministry of Education, Key Laboratory of Environment Friendly Management on High Altitude Rhododendron Diseases and Pests, Institutions of Higher Learning in Guizhou Province, School of Life Science, Guizhou Normal University, Guiyang, China, 5 College of Computer Science, Guiyang University, Guiyang, China, 6 Guizhou Caohai Wetland Ecosystem National Observation and Research Station, Guizhou Academy of Forestry Sciences, Guiyang, China, 7 Institute of Agricultural and biological engineering, Guizhou University, Guiyang, China, 8 School of International Education, Guizhou Normal University, Guiyang, China

☯ These authors contributed equally to this work.
* jy203c@gznu.edu.cn (GC); llx@gznu.edu.cn (LL); shentie@gznu.edu.cn (TS)

**Data Availability Statement:** All relevant data are within the manuscript and its Supporting Information files.

## Abstract

Tea ranks among the top three most beloved non-alcoholic beverages worldwide and boasts significant economic and health benefits. In addition to *Camellia sinensis* (L.) O. Kuntze, and other *Camellia* plants in China are consumed by residents as tea drinks, which also have important economic value. The present study introduces one of the wild tea species, namely, *Camellia tachangensis* F. C. Zhang. We analyzed changes in metabolite abundance and gene expression patterns of *C. tachangensis* and *C. sinensis* using metabonomics and transcriptomics. We found 1056 metabolites, including 256 differential metabolites (67 upregulated and 189 downregulated). Additionally, transcriptome analysis revealed 8049 differentially expressed genes, with 4418 upregulated and 3631 downregulated genes. *C. sinensis* boasts a notable abundance of Amino acids, which can be attributed to its specific genetic makeup. In Theanine and Caffeine metabolic pathways, the levels of the majority of amino acids and caffeine tend to decrease. In Flavonoid biosynthesis, the levels of the Flavanone Fustin and Epicatechin are higher in *C. tachangensis*, while Epigallocatechin and Gallocatechin levels are higher in *C. sinensis*. This indicates that the metabolic components of *C. sinensis* and *C. tachangensis* are not identical, which may result in a unique flavor.

**Funding:** This work was supported by Guizhou Provincial Science and Technology Projects [QIANKEHEJICHU-ZK [2021] Key 038, QIANKEHEZHICHENG [2022] Key 017, QIANKEHEPINGTAIRENCAI [2017] 5726-15 ], Guizhou Provincial Basic Research Program (Natural Science) [QIANKEHEJICHU-ZK [2023] 268], National Science Foundation of China NSFC [32260225], Natural Science Foundation of China and the Karst Science Research Center of Guizhou Province, China (U1812401), Key Laboratory of Environment Friendly Management on Alpine Rhododendron Diseases, Pests of Institutions of Higher Learning in Guizhou Province, Guizhou Normal University (Qianjiaoji[2022]044) and Coupling of Water and Fertilizer in Karst Desertification and Restoration of Biodiversity (Qian Jiao Ji [2023] No. 004). The funders had no role in study design, data collection and analysis, decision to publish, or preparation of the manuscript.

**Competing interests:** The authors declare that there are no conflicts of interest.

## Introduction

Tea ranks among the top three most beloved non-alcoholic beverages worldwide [1–3]. Tea has high nutritional and medicinal value, and it plays a crucial role in reducing atherosclerosis, vascular disease and stimulating the nerve surface [4]. *Camellia* belongs to the Theaceae family [5], *Camellia sinensis* is a type of *camellia* used for tea cultivation [6]. There are numerous varieties of tea plants worldwide, and China has over 3000 varieties of tea germplasm [7]. Yunnan-Guizhou Plateau is among the origin centers of tea. Because of its unique geology, diverse climate, and abundant precipitation within this area, the tea population's variety is being safeguarded [8], and there are abundant wild tea trees and modern local species with different forms [9]. In recent years, some high-quality tea resources discovered in the Yunnan-Guizhou Plateau have important academic research value and utilization potential, for example, Guizhou Niaowang tea [10], *Camellia tachangensis* F. C. Zhang [11], etc. *C. tachangensis* F. C. Zhang and *C.sinensis* both belong to the genus *Camellia* [12]. These species are widely distributed in the Yunnan-Guizhou Plateau and are used as substitute tea drinks by many local residents [12]. Tea contains abundant primary metabolites such as alkaloids [13], tea polyphenols [14], and theanine [15], as well as a rich variety of secondary metabolites [16], including flavonoids [17] and volatile compounds [18], which add to the distinct taste attributes of tea [19]. Tea's distinctive taste and aroma are derived from its primary metabolites (alkaloids, tea polyphenols, and theanine) and secondary metabolites (flavonoids and volatile compounds), which give it a delightful character as a beverage [20]. The important biochemical components in tea include caffeine, theanine, and tea polyphenols [21]. Caffeine, theanine, and tea polyphenols collectively account for about 3%, 1–2%, and 20–30% of the weight of dried tea leaves, as reported by Su, Wu, Wan, & Ning (2019) [22]. These natural components play key roles in shaping the distinct flavors of tea, including its bitterness, umami, and astringency [22]. Studies indicate that compounds like catechins, flavonoid-3-glycosides, caffeine, and theaflavic acid are believed to be responsible for the bitterness and astringency of tea, whereas amino acids such as glutamic acid, theanine, succinic acid, and gallic acid are thought to enhance the pleasant flavor of tea [15, 23].

Transcriptomic and metabolomic correlation analysis is a commonly employed method in investigating the abundance of plant metabolites [24, 25]. During the analysis of *C. tachangensis*, researchers delved into the exploration of the entire chloroplast genome sequence of this species [26]. The biosynthetic mechanism of proanthocyanidins in tea remains unclear. Previous studies have utilized high-performance liquid chromatography (HPLC) and liquid chromatography with diode array detection combined with triple quadrupole mass spectrometry (LC-DAD-QQQ). Researchers have found that proanthocyanidin dimers are abundant in *C. sinensis*, *C. assamica*, *C. taliensis*, *C. gymnogyna*, and *C. tachangensis* [12]. Using correlation analysis, an investigation was conducted to explore the correlation between sugar levels and proanthocyanidin dimers in five different types of tea, revealing a close relationship between glucose levels and proanthocyanidin dimer content. This provides theoretical data for the study of polyphenolic components and proanthocyanidin biosynthesis in certain species of tea [12]. Research has been conducted on the purple phenomenon exhibited by the tea tree variety *C. tachangensis* due to the increase in flavonoids and anthocyanins, providing an important foundation for the development of unique genetic resources and breeding [16]. Nonetheless, data are scarce concerning the taste characteristics of *C. tachangensis*, with a notable absence of a comprehensive and trustworthy scientific foundation. This gap hinders the exploration of the physiological and biochemical traits of *Camellia* species in the Yunnan-Guizhou Plateau, consequently stalling the progress and exploitation of *C. tachangensis*.

In the study currently being conducted, *C. tachangensis* was used as the test material, and *C. sinensis* as the control. This study conducted a comparative analysis of the differences in metabolite composition and content among different tea tree varieties, including flavonoids and other bioactive compounds such as phenolic acids and alkaloids. Exploring the similarities and differences in metabolic characteri stics between Guizhou tea plants (*C. tachangensis*) and *C. sinensis* can enhance our understanding of the unique metabolic pathways and traits of Guizhou tea trees. This helps us identify specific compounds that contribute to the distinctive flavor, aroma, and potential health benefits of Guizhou tea, providing foundational data for the development of tea beverage alternatives. This provides a foundation for subsequent research to identify the unique flavors, aromas, and potential health benefits of Guizhou tea, aiding in the development of alternative tea beverages.

## Materials and methods

The specimens under experimentation, *C. tachangensis* and *C. sinensis*, were initially planted in Pu'a County, Guizhou Province, situated in the southwestern region of China. These specimens were then collected from the Guizhou University Tea Science Experimental Base in Guiyang City, Guizhou Province, China. The coordinates for Pu'a County are N: 25°29'11", E: 104°59'10", while the experimental base in Guiyang City has coordinates N: 26°25'50", E: 106°40'47". In October 2021, fresh tea leaves of *C. tachangensis* and *C. sinensis* were each picked as experimental materials, with three samples taken in parallel for each type of tea, establishing three biological replicates. A total of six samples were collected from the two for *Camellia* species.

### RNA sequence analysis

This study utilizes the RNAprep Pure Plant Kit (product number: DP441) to fextract RNA from two types of tea leaves. The tea leaves are ground into powder using liquid nitrogen, then mixed with lysis buffer SL and centrifuged. The supernatant is transferred to column CS, followed by ethanol precipitation and transfer to column CR3 for centrifugation. Protein removal buffer RW1 is added, followed by DNase I treatment at room temperature for 15 minutes, and another round of protein removal buffer RW1 and centrifugation. Subsequent steps include adding washing buffer RW, centrifugation, and repeating the process. Finally, RNA is eluted from column CR3 with RNase-Free ddH2O, centrifuged after 2 minutes at room temperature to obtain RNA solution. The RNA samples are stored at -70°C. The total RNA samples were assessed for their purity, concentration, and integrity. The cDNA library was created utilizing sequencing by synthesis (SBS) technology by a leading biotech company in Beijing, China, known as Beijing Biomaker Biotech Co., Ltd. The NEBNext®Ultra™ RNA Library Prep Kit for Illumina® (NEB, USA) was employed to create the sequencing library, with an index code incorporated into the sequence attributes of every sample. Trinity (v 2.5.1) was employed to assemble clean data to acquire unigenes. The DIAMOND (version 2.0.4) program was employed to conduct a comparative analysis of the unigene sequence against the NR, Swiss-Prot, KEGG, and using KOBAS(V3.0)to obtain the KEGG Orthology results for Unigene, and analyzing the GO Orthology results of new genes with InterProScan(5.34–73.0) utilizing the integrated database of InterPro, after predicting the amino acid sequences of Unigene, the HMMER (V3.1B2) software was used to compare with the Pfam database to obtain annotation information for Unigene. The sequencing data was matched against the Unigene database using Bowtie. Then the assessment of gene expression levels was conducted by integrating the comparison outcomes with RNA-Seq through Expectation-Maximization (v 1.2.19) (RSEM). The quantification of gene expression levels is based on the Fragments Per Kilobase of

transcript per Million mapped reads (FPKM) value. A comparison of gene expression levels among samples from distinct groups was performed utilizing the DESeq2(v1.6.3) software. The criteria for identifying differentially expressed genes were a false discovery rate (FDR) of less than 0.01 and a minimum fold change of 2. The KOBAS software version 3.0 was employed for assessing the enrichment outcomes of distinctively expressed genes within the KEGG pathway [27–31].

## Metabolome sample extraction

The specimens were freeze-dried in a lyophilizer (Scientz-100F, made in Japan), and then ground using a grinder (MM 400, manufactured by Retsch in Germany). Subsequently, a precise 100 mg of powder was weighed, dissolved in 1.2 mL of a potent 70% methanol extraction solution, and left to rest at a chilly 4˚C overnight [32].

## UPLC–MS/MS conditions

The equipment utilized for data collection consisted of liquid chromatography technology (SHIMADZU Nexera X2 from Japan) paired with tandem mass spectrometry technology (Applied Biosystems QTRAP). Additionally, a high-quality Agilent SB-C18 column (1.8 μm, 2.1 mm *100 mm) was employed in the process. A consisted of ultra-pure water with a 0.1% formic acid concentration, whereas mobile phase B was a blend of acetonitrile infused with 0.1% formic acid. The elution profile used in the experiment was as described: Initially, the mobile phase consisted of 5% phase B at the start of the analysis. Over 9 minutes, there was a gradual increase in the proportion of phase B until it reached 95%. This composition was then held constant for an additional minute, from the 10th to the 11th minute. Subsequently, there was a rapid reduction in the concentration of phase B back to 5%, where it was maintained for the remaining 14 minutes of the elution process. The experimental conditions were set as follows: a rate of 0.35 milliliters per minute, a temperature setting of 40 degrees Celsius for the column, and an injection with a volume of 4 microliters were employed for the analysis. The LIT and triple quadrupole (QQQ) data were generated using a state-of-the-art triple quadrupole linear ion trap mass spectrometer (Q TRAP; AB6500 Q TRAP) UPLC/MS/MS system. This setup was equipped with an ESI turbo ion spray interface and was managed by Analyst 1.6.3 software, enabling functionality in both positive and negative ionization modes. The operational parameters for the electron spray ionization (ESI) source were set as follows: utilizing a turbo spray ion source, maintaining a source temperature of 550˚C, and applying an ion spray voltage (IS) of 5500 V in positive mode and -4500 V in negative mode. The ion source gases GSI and GSII were adjusted to 50 and 60 psi, while the curtain gas CUR was set at 25 psi, with the collision-induced ionization parameter dialed up to maximum intensity [33]. Calibration of instruments and adjustment of mass was conducted utilizing polypropylene glycol solutions at concentrations of 10 μmol/L in QQQ mode and 100 μmol/L in LIT mode. QQQ analysis was conducted utilizing the multiple reaction monitoring (MRM) mode and employing a moderate setting for the collision gas. Additionally, individual optimizations for declustering potential (DP) and collision energy (CE) were carried out for every MRM ion pair [34, 35]. Different MRM ion pairs were closely observed at various intervals according to the metabolites that were detected during each specific time frame. UPLC–MS/MS analysis was performed by BioMax Biotech (Beijing, China) [36].

## Qualitative and quantitative analysis

Metabolite quantification was carried out through the use of triple quadrupole mass spectrometry and Multiple reaction Monitoring (MRM) analysis. During the MRM process, the

quadrupole first sifts through the precursor ions of the desired substance, filtering out ions related to different molecular weights to effectively reduce potential interference. The precursor ions are fragmented into numerous ions upon activation and ionization within the collision chamber. Following this, the fragment ions produced are filtered by the triple quadrupole to separate a distinct characteristic ion, effectively eliminating any undesired interference from non-target ions. This process enhances the precision of quantification and improves the overall reproducibility of the analysis [37]. When the mass spectrometry data for metabolites in various samples had been collected, the peak areas of all substances were consolidated. Subsequently, Merge and adjust mass spectrometry peaks of the same metabolite from different samples [38]. The data from the mass spectrum were analyzed with the assistance of Analyst software [39].

Metabolite information was standardized for further statistical evaluation. orthogonal partial least squares discriminant analysis (OPLS-DA), principal component analysis (PCA), and Hierarchical clustering analysis (HCA) were conducted on the metabolite data of nine samples utilizing the R software version 3.1.1 to investigate the patterns of metabolite accumulation within the samples [40].

## Results

### Classification of metabolites

To gain a deeper insight into the alteration pattern of metabolites across various types of tea, the metabolites from *C. tachangensis* and *C. sinensis* were distinguished through the UPLC-MS/MS technology. A total of 1056 metabolites were identified in the analysis: 99 lipids; 72 alkaloids; 172 phenolic acids; 15 terpenes; 63 organic acids; 58 nucleotides and their derivatives; 301 flavonoids; 100 other metabolites; 52 lignins and coumarins; 35 tannins; and 89 amino acids and their derivatives (Table 1). In *C. tachangensis*, 1014 metabolites were uncovered, whereas in *C. sinensis*, a whopping 1039 metabolites were observed, revealing distinct metabolic characteristics within the specimens.

### Overview of metabolomics results

The findings from the principal component analysis indicated distinct segregation among the various species. As shown in Fig 1B, the first principal component (PC1) accounted for a significant 69.29% while the second principal component (PC2) elucidated 13.92% of the original

**Table 1. Overview of annotated metabolites.**

| Type | Number | Percentage (%) |
|---|---|---|
| Flavonoids | 301 | 28.5 |
| Phenolic acids | 172 | 16.3 |
| Others | 100 | 9.5 |
| Lipids | 99 | 9.4 |
| Amino acids and derivatives | 89 | 8.4 |
| Alkaloids | 72 | 6.8 |
| Organic acids | 63 | 6.0 |
| Nucleotides and derivatives | 58 | 5.5 |
| Lignans and Coumarins | 52 | 4.9 |
| Tannins | 35 | 3.3 |

Note: Primary Classification of Metabolites.

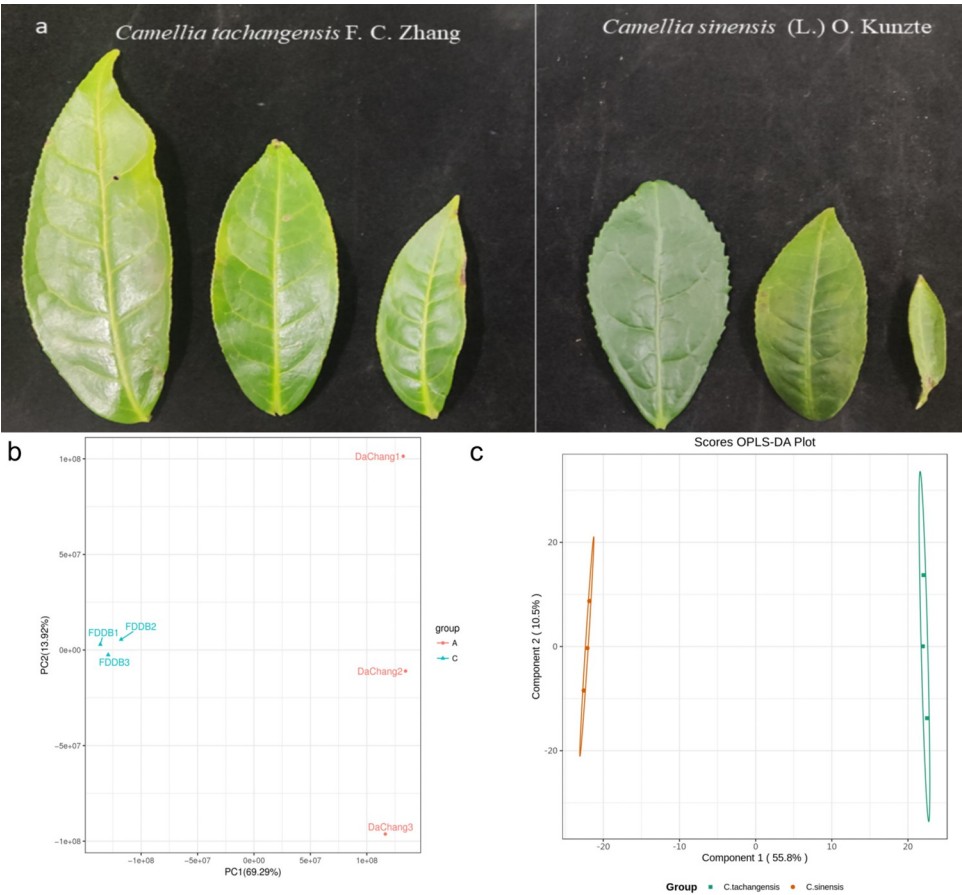

**Fig 1.** (a) Photos of leaves of *C. tachangensis* and *C. sinensis*. (b) PCA plot. DaCchang represents *C. tachangensis*, and FDDB represents *C. sinensis*. (c) OPLS-DA score plot. Every data point on the graph symbolizes a sample, with samples belonging to the identical group being depicted by the same color.

dataset's attributes. Fig 1B shows a significant overall difference between the samples of *C. tachangensis* and *C. sinensis*. In Fig 1C, the horizontal axis represents the predicted principal components, so the direction of the horizontal axis can show the differences between groups. There are notable distinctions between the samples of *C. tachangensis* and *C. sinensis* (Fig 1C), consistent with the PCA plot. The orthogonal principal components are depicted on the vertical axis, with the orientation of this axis revealing the variations among different groups.

## Differential metabolite analysis

Identified differential metabolites were filtered using fold change, VIP score, and P value as additional criteria. Metabolites meeting fold change (FC) $\geq 1$, VIP $\geq 1$, and P value $< 0.05$ simultaneously were regarded as differential metabolites. The findings from the screening are displayed in Fig 2A. In total, 256 different metabolites were identified in the screening process conducted on *C. tachangensis* and *C. sinensis*, including 67 upregulated metabolites and 189 downregulated metabolites. The first five metabolites qualitatively identified were selected and labelled in the figure after sorting according to the P value as follows: apigenin-6-C-(2"-glucosyl) arabinoside, N-feruloylagmatine 4-(3,4,5-trihydroxybenzyl) benzoic acid, galloylprocyanidin B4, and epigallocatechin (EGC). Fig 2B shows the differential metabolic cluster heatmap.

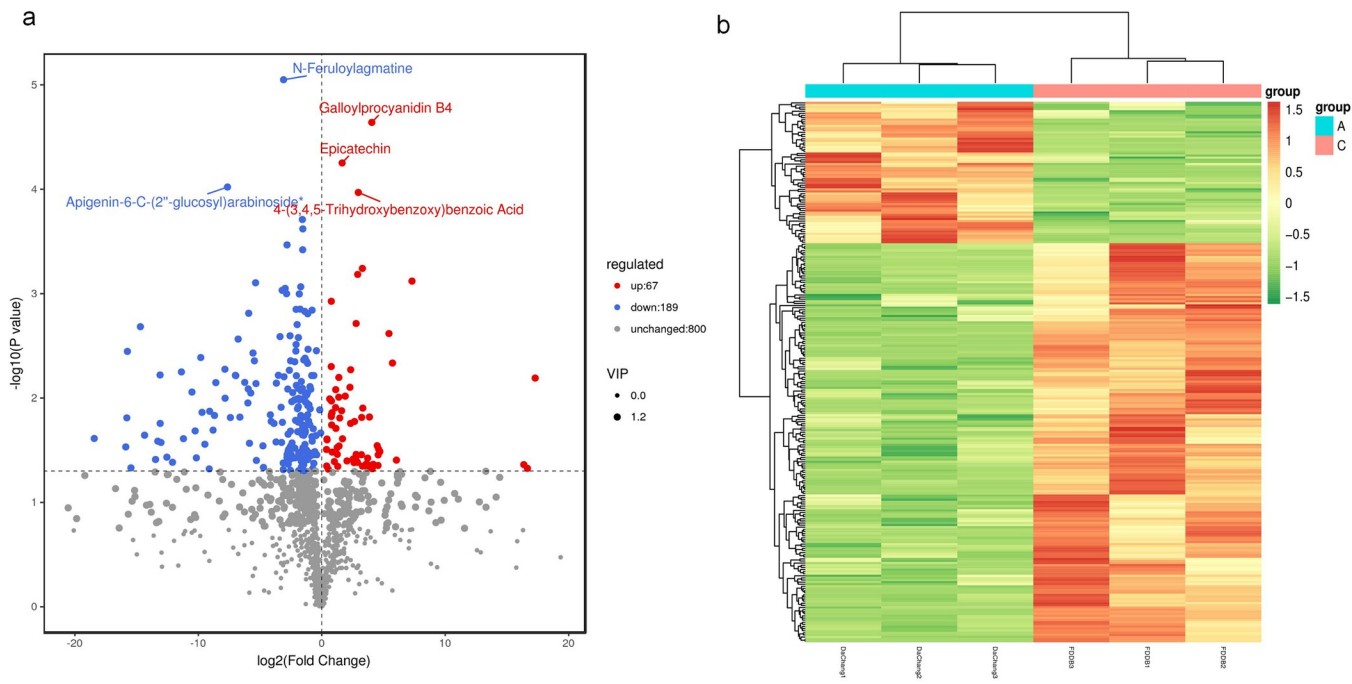

**Fig 2.** (a) Volcanic map of differential metabolites. (b)Differential metabolism cluster thermogram.

Differential metabolites were annotated to the HMDB database (Fig 3A), and it was found that the top 4 classes of differential metabolites with a higher number of annotations within this comparison group in the HMDB database were carboxylic acids and derivatives, fatty acyls, benzene and substituted derivatives, and flavonoids. The metabolites from *C. tachangensis* and *C. sinensis* were analyzed using the KEGG database to obtain detailed pathway data for the metabolites [38]. Enrichment analysis was carried out on the annotated findings to identify pathways containing enriched differential metabolites. The amino acid biosynthesis pathway, ABC transporter pathway, and glycine, serine, and threonine metabolic pathways are the main enriched pathways of differential metabolites (Fig 3A). Upstream substances were synthesized in the biosynthetic pathway of amino acids, which determined the material basis for the synthesis of secondary metabolites as indicated by the enrichment network diagram in Fig 3B. This is consistent with the results annotated to the HMDB database, with the most enriched class being carboxylic acids and derivatives, which include amino acid substances. These metabolic pathways were subsequently analyzed. KEGG significantly enriches the pathways of Biosynthesis of amino acids, ABC transporters, Lysine biosynthesis, Porphyrin and chlorophyll metabolism, and Glycine, serine and metabolism, with the metabolites associated with these pathways illustrated in Fig 3C. The substances significantly enriched in these pathways are mostly amino acid-like substances. Among which, amino acid-like substances such as Lysine, Proline, Threonine, Asparagine, etc., can be produced by the tricarboxylic acid cycle.

## KEGG enrichment analysis of metabolites

To further study the molecular mechanism of the quality difference between *C. tachangensis* and C. sinensis, RNA-seq analysis was performed. Extract RNA samples from these two tea plant species and perform sequencing analysis. Six libraries were constructed for the two varieties (with 3 biological replicates for each variety), resulting in 38.9 clean data. Each sample has 6.03Gb of clean data, with a Q30 base percentage of over 93.65%, and the GC content of each

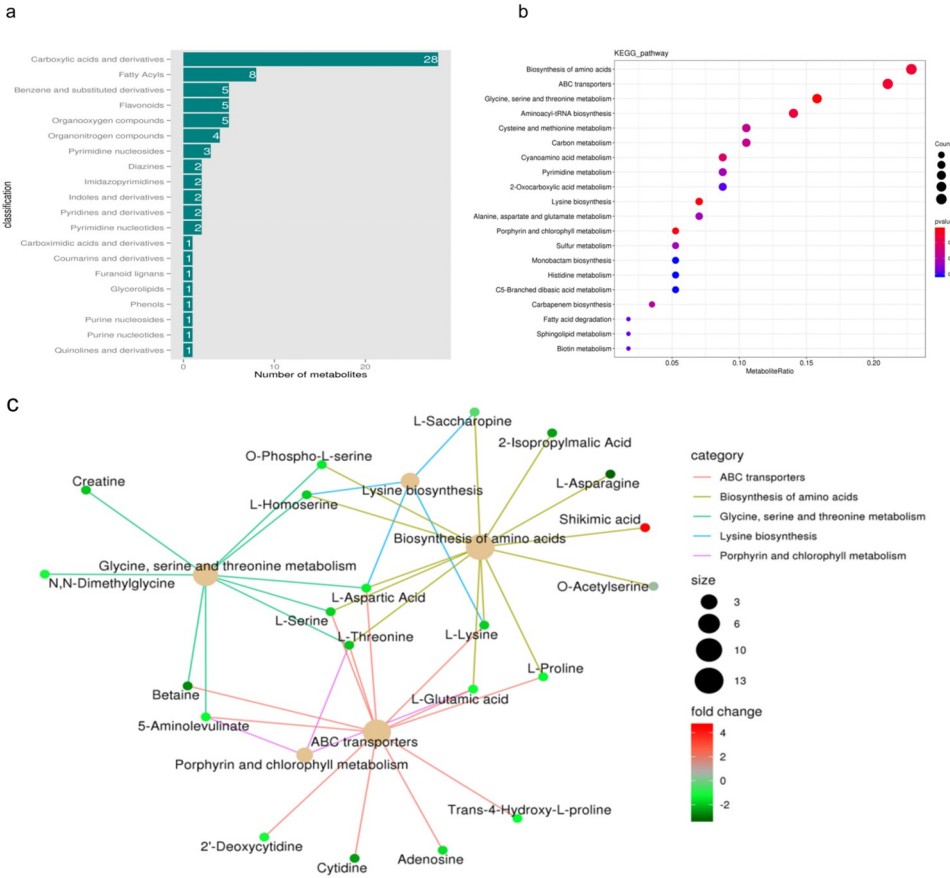

**Fig 3.** (a)Differential metabolite HMDB classification, (b) The KEGG enrichment plot of different metabolites, (c) Differential metabolite KEGG enrichment network diagram.

sample ranges from 44.94% to 45.51% (S1 Table). A total of 52,640 unigenes were obtained, of which 28,127 were longer than 1 kilobyte in size. The unigenes had an N50 of 1,907 and high assembly integrity (S2 Table). The unigenes were subjected to functional annotation analysis compared with various databases such as NR, Swiss-Prot, KEGG, COG, KOG, GO, and Pfam. This comprehensive analysis led to the identification of 35,323 annotation outcomes for the unigenes. An examination of gene composition utilizing the unigene repository was conducted, yielding a total of 18,897 SSR markers identified through SSR analysis(S3 and S4 Tables).

## Screening and functional annotation of differentially expressed genes

In the screening process of differentially expressed genes, an FDR < 0.01 and an FC ≥ 12 were employed as the criteria for filtering out differentially expressed genes. The comparison between *C. sinensis* and *C. tachangensis* revealed that there are 8,049 genes with differential expression, among which 4,418 genes showed increased expression and 3,631 genes showed decreased expression (Fig 4A). All differentially expressed genes were aligned with the GO database and subsequently sorted into three distinct groups: cellular components, molecular functions, and biological processes. In the biological process category, the genes that exhibited differential expression were predominantly focused on metabolic processes, cellular processes, and single-organism processes. In the cellular component category, the DEGs were mainly

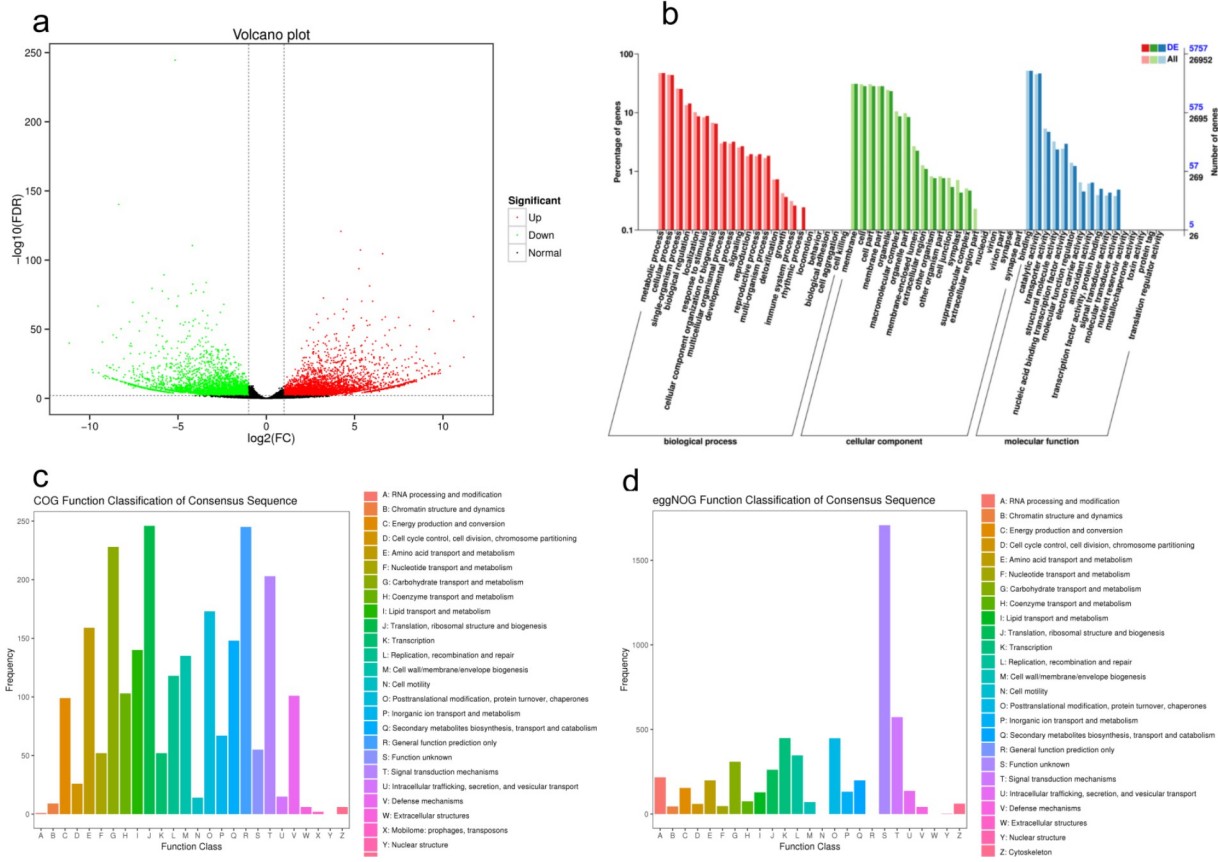

**Fig 4.** (a) Differential gene scatter plot. (b) GO second-level node annotation statistics of differentially expressed genes. (c) Differential gene COG annotation classification statistics chart. (d) Differential gene eggNOG annotation classification statistics chart.

enriched in the cell membrane, cell, cell part, and membrane part, while in the molecular function category, the DEGs were mainly enriched in binding, catalytic activity, and transporter activity (Fig 4B). Among the 7095 differentially expressed genes, 2091 single genes were classified into 25 functional orthologous groups (COG) clusters (Fig 4C). Among which, the unigenes in the "translation, ribosomal structure and biogenesis" cluster related to biosynthesis and metabolism functions and the "General function prediction only" cluster were the largest groups, while only a few unigenes were classified as "Nuclear structure". Within the eggNOG repository, a grand total of 1865 genes exhibited differential expression, predominantly linked to the category of "Signal transduction mechanisms" within the cluster, apart from the bulk of genes with unidentified functions among the differentially expressed ones (as depicted in Fig 4D).

## KEGG annotation results of differentially expressed genes

In the KEGG annotation analysis, a comparison between the two varieties, *C. sinensis* and *C. tachangensis*, revealed that a total of 8,409 genes were enriched in 131 KEGG metabolic pathways. Among these, the top 20 pathways with the strongest enrichment significance (i.e., the lowest Q values) are presented (Fig 5A). The KEGG enrichment analysis indicates that the five pathways with the highest enrichment levels among the differentially expressed genes in *C. sinensis* and *C. tachangensis* are, flavonoid biosynthesis, phenylpropanoid biosynthesis,

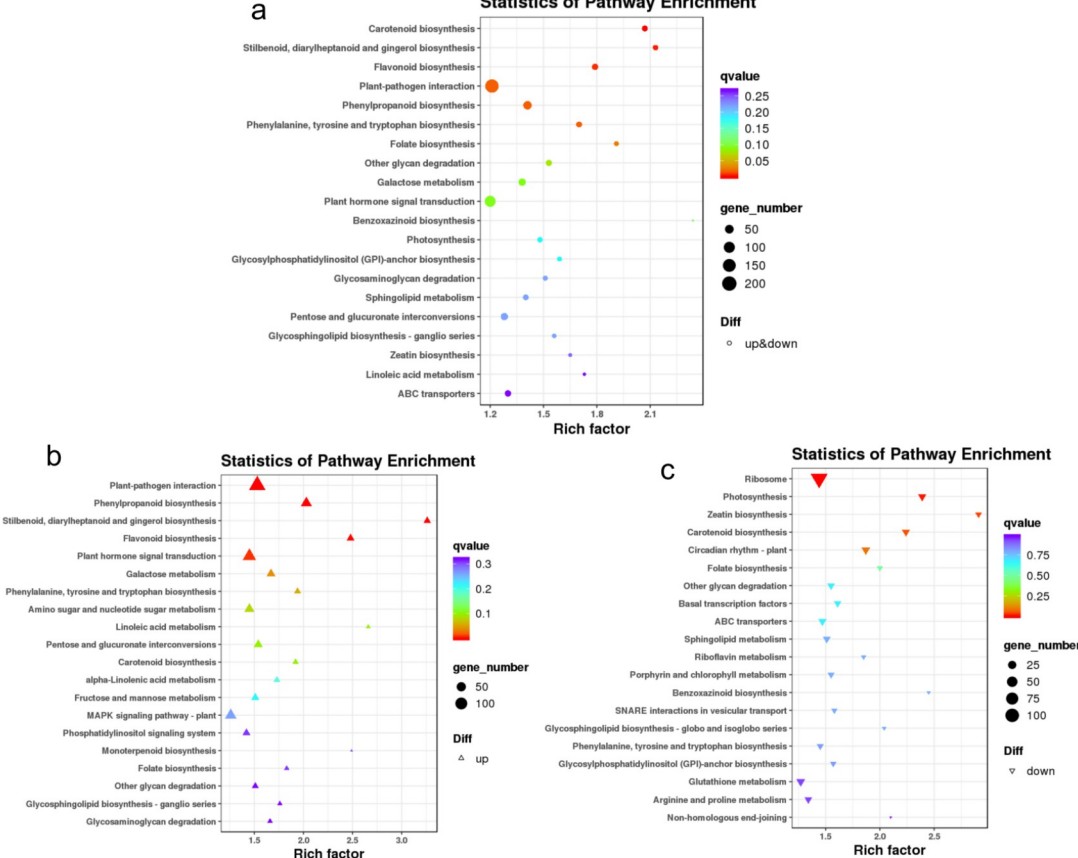

**Fig 5. KEGG enrichment bubble diagram of differentially expressed genes.** Scatter plot of KEGG pathway enrichment for all differentially expressed genes (DEGs) (a), up-regulated DEGs (b), and down-regulated DEGs (c).

styrene, diphenylheptane, and gingerol biosynthesis; phenylalanine, tyrosine, and tryptophan biosynthesis; as well as nucleotide metabolism related to the biosynthesis of phenylalanine, tyrosine, and tryptophan, which are associated with the flavor quality of tea. In Fig 5B, the majority of differentially expressed genes were significantly up-regulated in the stilbenoid, diterpenoid, and gingerol biosynthetic pathways, as well as in the phenylpropanoid and flavonoid biosynthetic pathways. The significantly enriched metabolic pathways of upregulated genes mostly belong to secondary metabolism synthesis pathways. In Fig 5C, the pathways of ribosome, photosynthesis, zeatin biosynthesis, and carotenoid biosynthesis show a notable enrichment of downregulated differentially expressed genes. The significantly enriched metabolic pathways of the latter two downregulated genes belong to terpenoid and polyketone compound pathways.

## Primary metabolism and gene coexpression networks in different tea species

To delve deeper into the primary elements contributing to the disparity in quality between *C. tachangensis* and *C. sinensis*, we mapped the coexpression network of primary metabolites and genes. As shown in Fig 6A and 6B, the metabolic pathways depicted are the theanine pathway and the caffeine pathway. In both the theanine and caffeine metabolic pathways, compared to *C. tachangensis*, most substances are down-regulated in *C. sinensis*. These data suggest that the

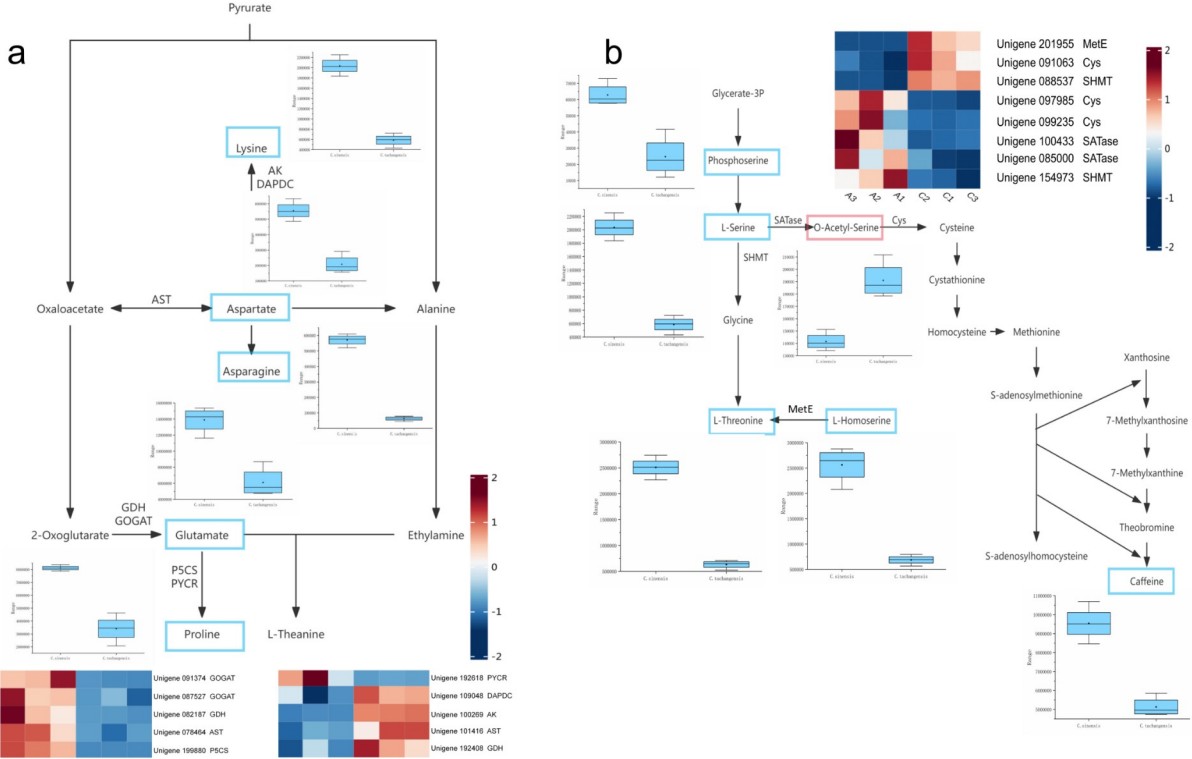

**Fig 6. Changes of major metabolites and related functional genes in *C. sinensis* and *C. tachangensis*.** (a) the catechin metabolism pathway. (b) the caffeine metabolism pathway. The elevated levels of differential metabolites are highlighted by the blue boxes in *C. sinensis*, while the red boxes indicate higher levels in *C. tachangensis*. C represents *C. sinensis*, and A represents *C. tachangensis*. The clustered graph shows the clustering of genes related to the enzymes marked in the pathways.

metabolic profiles of *C. tachangensis* and *C. sinensis* are not completely identical, which may contribute to the formation of unique flavors. These unique genes with significant differences in gene sequences may have a certain impact in terms of the build-up of major metabolites. The levels of serine, threonine, glutamate, proline, aspartate, and lysine metabolites in *C. sinensis* are all higher than in *C. tachangensis*. Compared to *C. sinensis*, the levels of most amino acids in *C. tachangensis* are significantly reduced, and the corresponding gene expressions of pyrroline-5-carboxylate reductase (PYCR), glutamate synthase (GOGAT), aspartate aminotransferase (AST) and pyrroline-5-carboxylate synthetase (P5CS) are downregulated. In the caffeine biosynthetic pathway, only the content of O-Acetyl-Serine is increased in *C. tachangensis*, and the upregulation of SATase-related genes may be the reason for this increase. The minimal build-up of free amino acids in *C. tachangensis* indicates a high level of nitrogen metabolism. The comparative relationship between enzymes and related genes in the caffeine and theanine metabolic pathways of *C. sinensis* and *C. tachangensis* can be seen in S5 Table.

*C. tachangensis* and *C. sinensis* showed different metabolic and gene regulation patterns in flavonoid biosynthesis (Fig 7). The comparative relationship between enzymes and related genes in the flavonoid metabolic pathways of *C. sinensis* and *C. tachangensis* can be seen in S6 Table. In *C. sinensis*, the levels of metabolites such as Epigallocatechin and Gallocatechin significantly increase. In *C. tachangensis*, metabolite levels, including the flavonol Fustin and Epicatechin, experience an increase. Within the flavonoid biosynthesis pathway, a notable increase in gene expression has been observed for the majority of enzyme-related genes, with particular emphasis on CHI, CHS, and CYP37A exhibiting pronounced upregulation. The

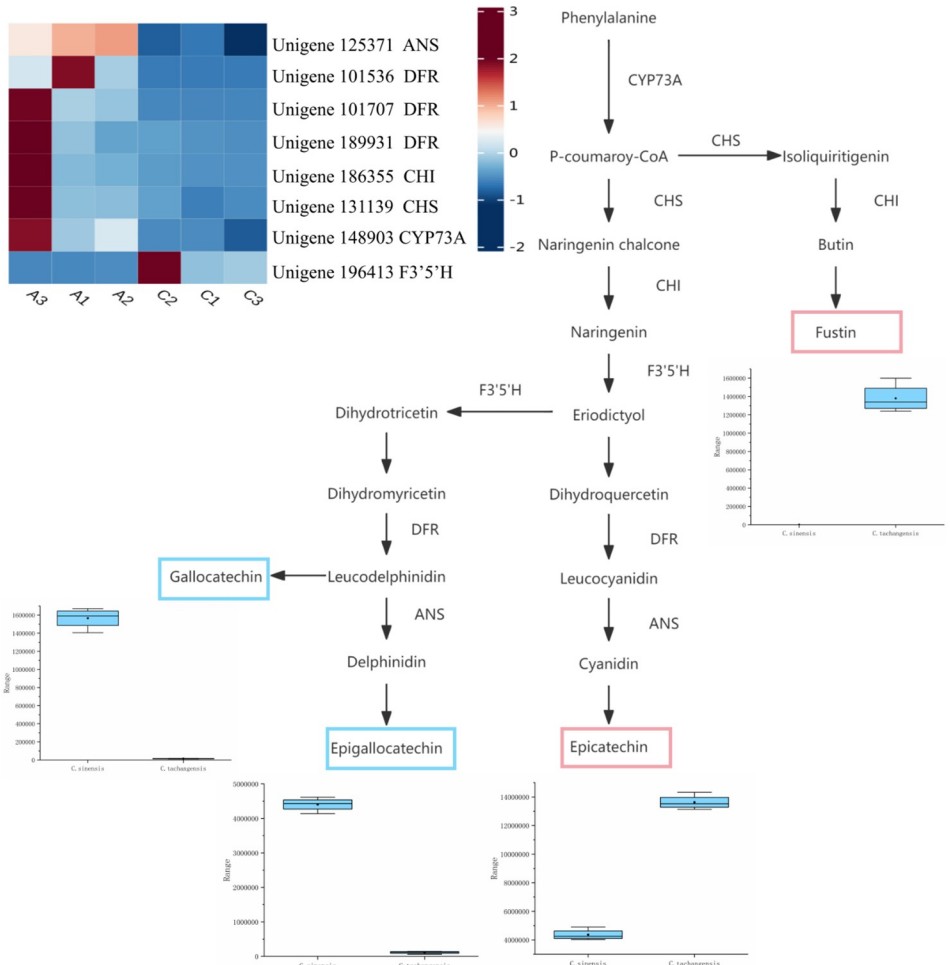

**Fig 7. Changes in the metabolic products of the flavonoid metabolic pathway and related functional genes in *C. sinensis* and *C. tachangensis*.** Blue boxes represent metabolites with higher levels in *C. sinensis*, while red boxes represent metabolites with higher levels in *C. tachangensis*. C represents *C. sinensis*, and A represents *C. tachangensis*. The clustered heatmap shows the clustering analysis of the related genes of the enzymes indicated in the pathway.

significant upregulation of these genes helps to synthesize upstream metabolites of the flavonoid biosynthetic pathway, creating favorable conditions for the biosynthesis of *flavonoid* compounds. The significant downregulation of F3'5'H-related genes may be associated with the downregulation of Epigallocatechin and gallocatechin expression. In our research, it was found that *C. sinensis* has a strong carbon metabolism capacity, which may be due to the accumulation of organic acids and the active expression of its biosynthetic enzyme genes. The differences in flavonoid metabolites between *C. sinensis* and *C. tachangensis* can be seen in S7 Table.

## Discussion

Research has found that the changes in key differential metabolites that affect tea quality are highly important. Many research findings have indicated that the primary factor that greatly impacts the quality of tea is caffeine, catechins, and theanine, with these flavor components being more prone to build up in the youthful leaves [41–43]. The level of astringency in tea is dictated by the concentration of catechins, while the bitterness is greatly influenced by caffeine,

and the freshness of tea is controlled by theanine [44, 45]. In the comparison between *C. tachangensis* and *C. sinensis*, 256 differential metabolites were successfully distinguished with 189 showing decreased levels and 67 showing increased levels. Upon closer examination, it was discovered that there was a significant increase in the levels of the metabolites Fustin, Epicatechin, and O-Acetyl-Serine in *C. tachangensis*, which were found to have a direct impact on the quality of tea leaves. In *C. sinensis*, the content of metabolites such as Lysine, Aspartate, Asparagine, Glutamate, Proline, L-Serine, L-Threonine, L-Homoserine, and Caffeine was found to be higher (Fig 6).

## Theanine and caffeine metabolic pathways

Theanine and caffeine play crucial roles in nitrogen metabolism within tea plants, standing out as the primary nitrogen-based compounds of significance. A study [46] compared *Camellia sinensis* (*C. sinensis*) with its most important relative, *Camellia taliensis*, and found that the caffeine content in *C. taliensis* is lower than that in *C. sinensis*. *C. sinensis* has been found to contain amino acids compared to *C. tachangensis*. Amino acids can enhance the delicate taste and distinctive flavor of tea, contributing to the sweetness, richness, and freshness of tea infusion [47]. Glutamic acid and aspartic acid contribute to the umami flavor, whereas proline imparts a hint of bitterness to tea [48, 49]. The concentrations of these three amino acids are higher in *C. sinensis* when compared to *C. tachangensis*. Qunfeng Zhang and colleagues [50] found that tea leaves exhibit a synergistic regulation over carbon and nitrogen metabolism, indicating that a decline in carbon metabolism can reduction in nitrogen [50]. In *C. tachangensis*, there is a reduction in the levels of the abundance of most tricarboxylic acid cycle metabolites. In *C. sinensis*, the abundance of most amino acid metabolites is increased. Previous studies have confirmed this observation, indicating that amino acids are the primary factors influencing the taste and functionality of *C. sinensis* [51]. After comparing *C. tachangensis* with *C. sinensis*, research has shown that C. tachangensis exhibits reduced levels of free amino acid build-up, indicating higher nitrogen metabolism. Research has shown that enhanced nitrogen catabolism provides more carbon skeletons for energy metabolism, resulting in the accumulation of organic acids [50]. Teaanine is a specific derivative of glutamic acid found in tea [52]. The composition of amino acids in tea leaves contributes to the taste of tea [53]. Additionally, the high expression of GOGAT and GDH in *C. sinensis* leads to the accumulation of upstream metabolite Glutamate in theanine metabolism (Fig 6), providing necessary conditions for the production of theanine. Glutamate dehydrogenase (GDH) is the upstream enzyme of glutamate and also a core enzyme in nitrogen metabolism [54]. Previous studies have shown that *C. sinensis* plants effectively assimilate ammonia [54]. Ammonia assimilation also occurs through the glutamate synthase cycle (GOGAT) [55]. In the comparison between *C. sinensis* and *C. tachangensis*, a decrease in glutamate content was discovered, and the related genes of GOGAT showed a down-regulated trend. However, the genes of GDH exhibited both up-regulated and down-regulated expressions. Key enzymes crucial for the synthesis of proline, P5CS, and PYCR, were also involved in this process [56, 57]. Studies have revealed that the P5CS gene shows heightened activity levels in response to cold temperatures [58]. PYCR stands as the ultimate pivotal enzyme in proline biosynthesis [57]. In the comparison between *C. sinensis* and *C. tachangensis*, the expression of PCRY and P5CS-related genes is downregulated, which may be the reason for the decrease in proline content. Research findings have shown that the activity of GOGAT significantly influences the theanine concentrations in tea leaves post-harvest [59]. In the process of generating lysine, the suppression of GOGAT-associated genes could potentially result in a decrease in lysine production.

## Catechins metabolic pathways

Catechins, as crucial elements in tea, are essential for providing flavor, while also offering beneficial properties such as antioxidants, antimicrobial effects, and potential anticancer properties [60–62]. Catechins found in tea leaves are crucial factors that contribute to the overall quality of tea [63]. Catechins and research have shown that they are largely responsible for the astringency and bitterness of tea [64]. A comparison of black tea, white tea, and green tea found that the content of catechin derivatives such as epicatechin gallate (ECG) and gallocatechin (GC) is higher in green tea extracts [65]. Previous studies have investigated multiple species of the *Camellia* genus, including *C. sinensis*, *C. assamica*, *C. taliensis*, *C. gymnogyna*, and *C. tachangensis* [12]. The fcontent of epicatechin (EC) in *C. tachangensis* is significantly higher than that in *C. sinensis*, which is consistent with the findings of this study. Additionally, the content of EC and epigallocatechin (EGC) in *C. tachangensis* is significantly higher than in *C. assamica* [25]. The comparison between *C. sinensis* and *Camellia taliensis* (*C. taliensis*) in the study [46] also revealed that the content of EGC, EC, and catechin gallate (GC) is higher in *C. sinensis* than in *C. taliensis*. In our study, we found that the EC content in *C. sinensis* is lower than in *C. tachangensis*. From this, it can be inferred that the EC content in *C. tachangensis* is higher than in *C. taliensis*. In *C. sinensis*, the higher levels of differential metabolites Epigallocatechin and gallocatechin (Fig 7) indicate that it has higher antioxidant activity compared to *C. tachangensis*. The differential metabolites with higher content in *C. tachangensis* are Epicatechin and Fustin. The data presented in Fig 7 indicates that the genes CYP73A, CHS, and CHI have all shown an increase in expression levels, potentially contributing to the rise in Fustin content. Research has shown that the expression of ANS results in the buildup of EGC [66]. DFR and ANS-related genes are both up-regulated, which potentially could be the underlying cause for the increase in Epicatechin levels. A recent study conducted by Jin and colleagues has revealed aallele gene in the F3'5'H gene that is linked to the catechin index characteristic in wild tea plants [63]. The potential suppression of genes associated with F3'5'H could impact the reduction in Epigallocatechin and Gallocatechin expression levels.

## Prospects

Over the past few years, there has been a growing trend in research towards unconventional tea options, including vine tea [67], Tibetan tea [68], and insect tea [69]. Caffeine, free amino acids, and other chemical components are major constituents of tea, contributing significantly to its flavor, functionality, and quality. The content of tea amino acids, in particular, plays a crucial role [70, 71]. Some previous studies [70, 71] have provided important information for the development of new tea beverages, including the regulation of compounds and specific genes related to taste and aroma, which may have a key role in the synthesis of aroma components [72]. In this study, using transcriptomics and metabolomics approaches, we explored the molecular mechanisms underlying the production of different metabolites during the growth process of unconventional tea (*C. tachangensis*) and traditional tea (*C. sinensis*), revealing the reasons for their quality differences. The changes in key differential metabolites have significant effects on the quality of tea leaves.This study identified partial molecular pathways related to tea quality, such as the tea amino acid metabolism pathway, caffeine metabolism pathway, and catechin metabolism pathway, providing possibilities for the development of new tea varieties in the future. In agricultural practice, enhancing tea quality can be achieved by optimizing amino acid metabolism and promoting the biosynthesis of flavonoid compounds. There is great hope in meeting the ever-changing demands of consumers for high-quality tea products, thereby promoting the development of the tea industry.

## Conclusion

Tea contains rich primary and secondary metabolites, such as alkaloids, flavonoids, tea polyphenols, theanine, and volatile compounds, which make up the unique taste profiles of tea. The metabolites in tea provide tea with a unique taste and flavour. Besides *C. sinensis*, other *Camellia* plants in China are consumed by local residents as substitute tea drinks, which also have important economic value. The present study introduces one of the tea substitutes, namely, *Camellia tachangensis* F. C. Zhang. In the pathways of catecholamine and caffeine synthesis, the levels of the majority of amino acids and caffeine tend to drop, which is related to the manifestation of their encoding genes. During the production process of flavonoids, the levels of flavanone Fustin and its derivatives such as Epicatechin significantly increase, while the expression of Epigallocatechin and gallocatechin is downregulated. These data indicated that the metabolic components of *C. tachangensis* and *C. sinensis* are not identical and may form a unique flavour. Enhancing tea quality in agricultural practices can be achieved by optimizing amino acid metabolism and promoting the biosynthesis of flavonoids.

## Supporting information

**S1 Table. Sample sequencing data evaluation statistical.**
(DOC)

**S2 Table. Assembly result statistics.**
(DOC)

**S3 Table. Unigene annotation statistics.**
(DOC)

**S4 Table. Statistics of SSR analysis results.**
(DOC)

**S5 Table. The relationship between enzymes and related genes in the caffeine and theanine metabolic pathway*s of C. sinensis and C. tachangensis.**
(DOC)

**S6 Table. The relationship between enzymes and related genes in the flavonoid metabolic pathways of *C. sinensis* and *C. tachangensis*.**
(DOC)

**S7 Table. Metabolite differences of *C. tachangensis* and *C. sinensis*.**
(DOC)

## Author Contributions

**Conceptualization:** Qihang Zhou, Xinglin Wang, Tie Shen.

**Data curation:** Xu Zhang.

**Formal analysis:** Zhengdong Zhang.

**Investigation:** Xingpan Meng.

**Methodology:** Guiping Chen, Lunxian Liu.

**Project administration:** Suzhen Niu.

**Supervision:** Ximin Zhang.

**Writing – original draft:** Yunfei Xu.

**Writing – review & editing:** Yunfei Xu, Qihang Zhou, Xinglin Wang.

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
