## [Decision Letter · Decision Letter 0]

26 Aug 2024

PONE-D-24-29681Metabonomics and Transcriptomics Analyses Reveal Quality Differences between Camellia tachangensis F. C. Zhang and C. sinensis (L.) O. KunztePLOS ONE

Dear Dr. Shen,

Thank you for submitting your manuscript to PLOS ONE. After careful consideration, we feel that it has merit but does not fully meet PLOS ONE’s publication criteria as it currently stands. Therefore, we invite you to submit a revised version of the manuscript that addresses the points raised during the review process.

We look forward to receiving your revised manuscript.

Kind regards,

Wajid Zaman

Academic Editor

PLOS ONE

Journal requirements: 1. When submitting your revision, we need you to address these additional requirements. Please ensure that your manuscript meets PLOS ONE's style requirements, including those for file naming. The PLOS ONE style templates can be found at https://journals.plos.org/plosone/s/file?id=wjVg/PLOSOne_formatting_sample_main_body.pdf and https://journals.plos.org/plosone/s/file?id=ba62/PLOSOne_formatting_sample_title_authors_affiliations.pdf. 2. Please note that PLOS ONE has specific guidelines on code sharing for submissions in which author-generated code underpins the findings in the manuscript. In these cases, all author-generated code must be made available without restrictions upon publication of the work. Please review our guidelines at https://journals.plos.org/plosone/s/materials-and-software-sharing#loc-sharing-code and ensure that your code is shared in a way that follows best practice and facilitates reproducibility and reuse. 3. In your Methods section, please provide additional information regarding the permits you obtained for the work. Please ensure you have included the full name of the authority that approved the field site access and, if no permits were required, a brief statement explaining why. 4. We note that the grant information you provided in the ‘Funding Information’ and ‘Financial Disclosure’ sections do not match.  When you resubmit, please ensure that you provide the correct grant numbers for the awards you received for your study in the ‘Funding Information’ section. 5. We note that Figure 1a in your submission contain copyrighted images. All PLOS content is published under the Creative Commons Attribution License (CC BY 4.0), which means that the manuscript, images, and Supporting Information files will be freely available online, and any third party is permitted to access, download, copy, distribute, and use these materials in any way, even commercially, with proper attribution. For more information, see our copyright guidelines: http://journals.plos.org/plosone/s/licenses-and-copyright. We require you to either (1) present written permission from the copyright holder to publish these figures specifically under the CC BY 4.0 license, or (2) remove the figures from your submission: a. You may seek permission from the original copyright holder of Figure 1a to publish the content specifically under the CC BY 4.0 license.  We recommend that you contact the original copyright holder with the Content Permission Form (http://journals.plos.org/plosone/s/file?id=7c09/content-permission-form.pdf) and the following text:“I request permission for the open-access journal PLOS ONE to publish XXX under the Creative Commons Attribution License (CCAL) CC BY 4.0 (http://creativecommons.org/licenses/by/4.0/). Please be aware that this license allows unrestricted use and distribution, even commercially, by third parties. Please reply and provide explicit written permission to publish XXX under a CC BY license and complete the attached form.” Please upload the completed Content Permission Form or other proof of granted permissions as an ""Other"" file with your submission.  In the figure caption of the copyrighted figure, please include the following text: “Reprinted from [ref] under a CC BY license, with permission from [name of publisher], original copyright [original copyright year].” b. If you are unable to obtain permission from the original copyright holder to publish these figures under the CC BY 4.0 license or if the copyright holder’s requirements are incompatible with the CC BY 4.0 license, please either i) remove the figure or ii) supply a replacement figure that complies with the CC BY 4.0 license. Please check copyright information on all replacement figures and update the figure caption with source information. If applicable, please specify in the figure caption text when a figure is similar but not identical to the original image and is therefore for illustrative purposes only.

Additional Editor Comments:

Dear authors,

Could you please double check about your work and wrtie in details that what is difference between the work already published by you (doi: 10.48130/BPR-2023-0003

Please clarify the novelty of the present study and how it was different from the previous one). Eventhough, the other study focus on flavonoids contents only.

Reviewers' comments:

Reviewer's Responses to Questions

**Comments to the Author**

1. Is the manuscript technically sound, and do the data support the conclusions?

Reviewer #1: Yes

Reviewer #2: Yes

2. Has the statistical analysis been performed appropriately and rigorously? 

Reviewer #1: Yes

Reviewer #2: I Don't Know

3. Have the authors made all data underlying the findings in their manuscript fully available?

Reviewer #1: Yes

Reviewer #2: Yes

4. Is the manuscript presented in an intelligible fashion and written in standard English?

Reviewer #1: No

Reviewer #2: Yes

5. Review Comments to the Author

Reviewer #1: In the title, the authors used the term “Metabonomics” but in rest of the manuscript, they used metabolome. Would be better to replace it with metabolome

The manuscript is full of typographic mistakes please carefully correct these mistakes.

Line 29: The genus name should be italicized.

Line 37: flavanone Fustin and Epicatechin? Why do some words start from small and some from capital letters?

Please provide a clear conclusion at the end of the abstract section.

Keywords: Why do the authors only mention Camellia tachangensis here?

46: Space problem.

48: Same as above.

50: Same as above.

51: Same as above.

Introduction

The first is about the general significance of tea. Please mention the family name of the Camellia first, and then mention the total number of species in the world and then mention the total number of species in China.

58: Space problem.

61: Space problem.

61: “and there are abundant wild tea trees” Which wild species of tea? Please mention the scientific name of the species as well.

“and modern local species” Please clarify what is meant by modern local species.

62-63: “In recent years, some high-quality tea resources discovered in the Yunnan-Guizhou Plateau have important academic research value and utilization potential.” Please mention their name as well.

64: “Zhang and Camellia sinensis ‘Fuding Dabaicha’ belongs to the Theaceae and Camellia genera C. tachangensis F. C.? Very confusing, please rewrite this sentence.

59-66: This should be part of the first paragraph.

Overall, the first paragraph should be about the total number of species in Camellia, family, then the number of species in this genus in China, distribution in the world, then in the study area, economic and medicinal potential,

67: Space problem

Line 84: No need to mention Figure 1 in the introduction part.

84-85. This should be the part of methods section. Here the authors only need to focus on the aims and objectives of the study.

Line 87: “To explore the similarities and differences in metabolic characteristics between Guizhou tea plants and C. sinensis” Why do the authors identify the differences between metabolites? Please provide a reason in the subsequent sentence.

93-66: This sentence does not make any sense. Please rewrite and convey a clear message.

99: for the two types of tea.? Or for two Camellia species?

120: Grammatical mistake.

156: MRM analysis, write the full name of MRM.

176: 1056 metabolites? It should be “A total of 1056 metabolites….”

108: “utilizing other databases…” Please specify the databases.

114: R (version 1.6.3).? but in line 170 the authors mentioned version 4.0.2?

183-198: “Would be better if the authors provides eigenvalue variance.percent cumulative.variance percent of the PC1, PC2 and PC3 in table form.

235-236. “RNA samples were extracted from a pair of distinct tea cultivars and subjected to sequencing analysis” provides the results of the PCR (band), and also specifies RNA extraction method in the Material and Methods Section.

250-251: In total, 8049 genes showed differential expression. In C.sinensis? or C. tachangensis.

271-291: The authors explain KEGG annotation results of differentially expressed genes but did not specify the species.

293: Why do the authors use the term cultivar? Please confirm the species name first with the World Flora Online and correct this all over the manuscript.

371: Follow the journal guidelines for reference format.

At the end of the discussion section please add a separate paragraph about the innovation and future prospects of the research.

Arrange the discussion section in a systematic as per the heading in the results.

No need to add citations in the conclusion section.

I am worried about the novelty of the manuscript as most of the work has already been published

doi: 10.48130/BPR-2023-0003

Please clarify the novelty of the present study and how it was different from the previous one.

Reviewer #2: The MS used the metabonomics and transcriptomics to tell the quality differences between Camellia tachangensis and C. sinensis.

There are several species and varieties for making tea, so the author is encouraged to discuss the samilarity and differences between different species of tea plants.

6. PLOS authors have the option to publish the peer review history of their article (what does this mean?). If published, this will include your full peer review and any attached files.

Reviewer #1: No

Reviewer #2: No

---

## [Author Response · Author response to Decision Letter 0]

23 Oct 2024

Some figure in our response on the system did not display correctly; you can view them in the attachments under 'Response to Reviewers'.

Responses to Academic Editorial Comments:

Dear Wajid Zaman,

Thank you for your email regarding the evaluation of my manuscript submission to PLOS ONE. I appreciate the feedback provided and the opportunity to revise the manuscript to meet the publication criteria of the journal.

Comment 1: Please include the following items when submitting your revised manuscript: 

Response 1: We have made the necessary changes to all submitted documents to comply with the style requirements of PLOS ONE, including the renaming of the files as requested.

You can see a detailed rebuttal letter, a marked-up copy of your manuscript, and an unmarked version of your revised paper in the system.

Comment 2: you would like to make changes to your financial disclosure, please include your updated statement in your cover letter. Guidelines for resubmitting your figure files are available below the reviewer comments at the end of this letter.

Response 2: We are providing the correct Financial Disclosure here. 

The correct version of "Funding Information" is as follows: 

“This work was supported by Guizhou Provincial Science and Technology Projects [QIANKEHEJICHU-ZK [2021] Key 038, QIANKEHEZHICHENG [2022] Key 017, QIANKEHEPINGTAIRENCAI [2017] 5726-15 ], Guizhou Provincial Basic Research Program(Natural Science) [QIANKEHEJICHU-ZK [2023] 268], National Science Foundation of China NSFC [32260225], Natural Science Foundation of China and the Karst Science Research Center of Guizhou Province, China (U1812401), Key Laboratory of Environment Friendly Management on Alpine Rhododendron Diseases, Pests of Institutions of Higher Learning in Guizhou Province, Guizhou Normal University (Qianjiaoji[2022]044) and Coupling of Water and Fertilizer in Karst Desertification and Restoration of Biodiversity (Qian Jiao Ji [2023] No. 004). The funders had no role in study design, data collection and analysis, decision to publish, or preparation of the manuscript.”

Thank you once again for the opportunity to revise the manuscript, and I look forward to resubmitting the revised version.

Best regards,

Tie Shen

Dear Reviewer,

We thank you for the opportunity to revise the manuscript “Metabolome and Transcriptomics Analyses Reveal Quality Differences between Camellia tachangensis F. C. Zhang and C. sinensis (L.) O. Kunzte” for publication in Plos one. We also thank the reviewers and you for the helpful comments. Following those suggestions, we have made several revisions to the manuscript as outlined below. We look forward to your response and hope the revisions will enable you to accept this manuscript version. In this revised version, changes to our manuscript were all highlighted within the document using red-colored text.

We also appreciate your clear and detailed feedback and hope the explanation fully addresses your concerns. In the remainder of this letter, we discuss your comments individually along with our corresponding responses.

To facilitate this discussion, we first retype your comments in italic font and then present our responses to the comments.

Responses to Additional Editor Comments:

Comment 1: Could you please double check about your work and wrtie in details that what is difference between the work already published by you (doi: 10.48130/BPR-2023-0003

Please clarify the novelty of the present study and how it was different from the previous one). Eventhough, the other study focus on flavonoids contents only. 

Response 1: In order to clarify the differences between the previously published work (DOI: 10.48130/BPR-2023-0003) and the current study, let us delve into specific aspects. The earlier publication focused on the analysis of flavonoid compound content, which is a crucial area in plant chemistry research. In contrast, this study not only expanded upon this foundation by analyzing a wider range of plant chemical substances but also integrated advanced analytical techniques to explore their synergistic effects.

The novelty of this study lies in its systematic analysis of plant chemistry, including metabolic pathways related to tea amino acids, caffeine, catechins, and other compounds associated with tea flavor. This multidimensional analysis not only provides a more comprehensive understanding of plant chemical components but also reveals potential interactions among these compounds that may impact their bioavailability and physiological effects. Additionally, this study employed advanced analytical techniques such as LC-MS/MS, which offer higher sensitivity and specificity compared to traditional methods used in previous work, enabling a more reliable and in-depth study of the precise quantification and characterization of plant chemical compositions.

While earlier research made significant contributions by focusing on flavonoid compounds in this field, this study expanded the breadth of plant chemical analysis. By utilizing metabolomics and transcriptomics methods, it analyzed changes in metabolite abundance and gene expression patterns in the metabolic pathways of tea amino acids, caffeine, and catechins in two plant species. This further extended the impact of the study. This approach not only strengthens the existing knowledge base but also opens up new avenues for exploring the therapeutic potential and applications of bioactive compounds derived from plants.

responses to Reviewers' comments:

Responses Reviewer #1

Comment 1: In the title, the authors used the term “Metabonomics” but in rest of the manuscript, they used metabolome. Would be better to replace it with metabolome

Response 1: Thank you for pointing out this issue. We also agree that it would be better to change "metabolome" to "metabolomics", and we have made the change throughout the article. You can check the title, lines1 and 140, and other locations for the updated term.

Comment 2: The manuscript is full of typographic mistakes please carefully correct these mistakes.

Line 29: The genus name should be italicized.

Response 2: Thank you for pointing out this issue. The attribution has been changed to italics. You can check the updated formatting in line 29.

Comment 3: Line 37: flavanone Fustin and Epicatechin? Why do some words start from small and some from capital letters?

Response 3: Apologies for our oversight that led to the incorrect capitalization of the words. We have now made the necessary changes, and you can review the corrected capitalization in lines 36 and 38.

Comment 4: Please provide a clear conclusion at the end of the abstract section.”

Response 4: Thank you for bringing up this issue.We have already provided additional information in the abstract section. You can refer to lines 36-37 and 39-41 for more details.

Comment 5: Keywords: Why do the authors only mention Camellia tachangensis here?

Response 5: Thank you for your query. The aim of this study is to explore the similarities and differences in the metabolic characteristics of Camellia tachangensis (C. tachangensis) and Camellia sinensis (C. sinensis) in Guizhou, China. C. tachangensis is the main focus of the research, while C. sinensis serves as a comparison. Therefore, only Camellia tachangensis is included in the keywords.

Comment 6-9: Space problem.

Response 6-9: Thank you for pointing out this issue. We have made changes to address the "Space problem" issue. You can review the revisions in lines 58, 60, 62, and 63.

Comment 10: The first is about the general significance of tea. Please mention the family name of the Camellia first, and then mention the total number of species in the world and then mention the total number of species in China.

Response 10: Thank you for your input. We have already made additions to this section. You can find it in lines 47-49.

Comment 11-12: Space problem.

Response 11-12: Thank you for pointing out this issue. . We have made changes to address the "Space problem" issue. You can review the revisions in lines 51 and 70.

Comment 13: 61“and there are abundant wild tea trees” Which wild species of tea? Please mention the scientific name of the species as well.

“and modern local species” Please clarify what is meant by modern local species.

Response 13: Thank you for your question. In the context of tea cultivation in the Yunnan-Guizhou Plateau, "modern local varieties" refer to tea tree cultivars developed or adapted to local conditions and requirements. These modern local species undergo selective breeding or hybridization to enhance certain characteristics such as flavor, yield, resistance to pests and diseases, and adaptability to the specific climate and soil conditions of the plateau.

These modern local varieties may have been developed through traditional breeding methods or modern agricultural techniques and are well-suited for tea cultivation in the Yunnan-Guizhou Plateau. These varieties may include cultivars specifically bred for certain flavor profiles or unique characteristics required in tea production.

Comment 14: 62-63: “In recent years, some high-quality tea resources discovered in the Yunnan-Guizhou Plateau have important academic research value and utilization potential.” Please mention their name as well

Response 14: Thank you for your input. We have supplemented some high-quality teas discovered in the Yunnan-Guizhou Plateau, such as Guizhou Niaowang tea, Camellia tachangensis F. C. Zhang, etc. You can refer to lines 54-55 for details.

Comment 15: 64: “Zhang and Camellia sinensis ‘Fuding Dabaicha’ belongs to the Theaceae and Camellia genera C. tachangensis F. C.? Very confusing, please rewrite this sentence.

Response 15: We sincerely apologize for the error that occurred. We have rewritten this sentence, and you can refer to lines 55-56 for the correct version.

Comment 16: 59-66: This should be part of the first paragraph.

Response 16: Thank you for your input. We have moved this section to the first paragraph. You can find it in lines 49-57.

Comment 17: Overall, the first paragraph should be about the total number of species in Camellia, family, then the number of species in this genus in China, distribution in the world, then in the study area, economic and medicinal potential

Response 17: Thank you for your feedback. We have made the requested changes in the first paragraph. You can see it in the first paragraph.

Comment 18: 67: Space problem

Response 18: Thank you for pointing out this issue. You can review the revisions in line 72.

Comment 19: Line 84: No need to mention Figure 1 in the introduction part.

Response 19: Thank you for your feedback. We have removed Figure 1 from the introduction.

Comment 20: 84-85. This should be the part of methods section. Here the authors only need to focus on the aims and objectives of the study.

Response 20: Thank you for your feedback. We have deleted this section from the introduction. You can see it in sections 91-93.

Comment 21: Line 87: “To explore the similarities and differences in metabolic characteristics between Guizhou tea plants and C. sinensis” Why do the authors identify the differences between metabolites? Please provide a reason in the subsequent sentence.

Response 21: We have provided the reasons in the original text, as follows: "This helps us identify specific compounds that contribute to the distinctive flavor, aroma, and potential health benefits of Guizhou tea, providing foundational data for the development of tea beverage alternatives. This provides a foundation for subsequent research to identify the unique flavors, aromas, and potential health benefits of Guizhou tea, aiding in the development of alternative tea beverages." You can refer to lines 93-100 in the original text for more details.

Comment 22: 93-96: This sentence does not make any sense. Please rewrite and convey a clear message

Response 22: Thank you for your suggestion. We have rewritten the sentence, and you can refer to lines 103-107 for the revised version.

Comment 23: 99: for the two types of tea.? Or for two Camellia species?

Response 23: We have made the correction to this error. It should be "for two Camellia species." You can review it in line 110.

Comment 24: 120: Grammatical mistake.

Response 24: Thank you for pointing out our mistake. We have made the correction to the grammar, and you can see it in lines 141-142.

Comment 25: 156: MRM analysis, write the full name of MRM.

Response 25: We have supplemented the full name, and you can see it in line 176.

Comment 26: 176: 1056 metabolites? It should be “A total of 1056 metabolites….”

Response 26: We have made the correction to this error. We have changed the error to "A total of 1056 metabolites." You can see it in line 195.

Comment 27: 108“utilizing other databases…” Please specify the databases.

Response 27: Thank you for pointing out our issue. We have supplemented the content from other databases, which you can review in lines 127-131.

Comment 28: 114: R (version 1.6.3).? but in line 170 the authors mentioned version 4.0.2?

Response 28: Thank you for pointing out our issues. In the article, version 1.6.3 refers to the DESeq2 software in the R package. To avoid ambiguity, we have made modifications to indicate DESeq2 (v1.6.3). The version of R stated as 170 was incorrect due to an oversight in translation; it has been corrected to version 3.1.1. These two R packages are used for gene analysis and metabolite analysis, respectively. You can review this in lines 136 and 190.

Comment 29: 183-198: “Would be better if the authors provides eigenvalue variance.percent cumulative.variance percent of the PC1, PC2 and PC3 in table form.

Response 29: Thank you for your suggestion. In this study, we first used PCA for data dimensionality reduction and visualization, observing the distribution patterns between samples. Then, we combined OPLS-DA score plots to further investigate the differences between different sample groups and identify the variables that primarily contribute to these differences. These two types of plots complement each other and provide a comprehensive understanding and interpretation of the dataset. We provide a 3D plot of PCA here, where PC1 accounts for 69.29%, PC2for 13.92%, and PC3 for 9.18%.

Please see the attachments for this part of the response and the figures.

fig. PCA 3D plot. DaCchang represents C. tachangensis, and FDDB represents C. sinensis.

Comment 30: 235-236. “RNA samples were extracted from a pair of distinct tea cultivars and subjected to sequencing analysis” provides the results of the PCR (band), and also specifies RNA extraction method in the Material and Methods Section.

Response 30: Thank you for your suggestion. We have now incorporated the RNA extraction method into the Materials and Methods section for reference. You can find it in lines 112-120. The following data and detection conclusions are based on the sequencing sample testing standards required by Beijing Biomarker. They provide a comprehensive evaluation of the tested samples. As shown in the table, all our sample detection results are categorized as A, indicating compliance with the library construction requirements in terms of quality and meeting the standard quantity for library construction of 2 times or more. Additionally, we have provided peak graphs of RNA detection in the subsequent section, which can help assess the sample quality.

Please see the attachments for this part of the response and the figures. 

Comment 31: 250-251: In total, 8049 genes showed differential expression. In C.sinensis? or C. tachangensis.

Response 31: Thank you for pointing out our issue. We compared the genes of C. sinensis and C. tachangensis and found 8049 differentially expressed genes between them. You can review it from line 

---

## [Decision Letter · Decision Letter 1]

30 Oct 2024

PONE-D-24-29681R1Metabolome and Transcriptomics Analyses Reveal Quality Differences between Camellia tachangensis F. C. Zhang and C. sinensis (L.) O. KunztePLOS ONE

Dear Dr. Shen,

Thank you for submitting your manuscript to PLOS ONE. After careful consideration, we feel that it has merit but does not fully meet PLOS ONE’s publication criteria as it currently stands. Therefore, we invite you to submit a revised version of the manuscript that addresses the points raised during the review process.

 Please submit your revised manuscript by Dec 14 2024 11:59PM. If you will need more time than this to complete your revisions, please reply to this message or contact the journal office at plosone@plos.org. Please include the following items when submitting your revised manuscript:A rebuttal letter that responds to each point raised by the academic editor and reviewer(s). You should upload this letter as a separate file labeled 'Response to Reviewers'.A marked-up copy of your manuscript that highlights changes made to the original version. You should upload this as a separate file labeled 'Revised Manuscript with Track Changes'.An unmarked version of your revised paper without tracked changes. You should upload this as a separate file labeled 'Manuscript'.If applicable, we recommend that you deposit your laboratory protocols in protocols.io to enhance the reproducibility of your results. Protocols.io assigns your protocol its own identifier (DOI) so that it can be cited independently in the future. For instructions see: https://journals.plos.org/plosone/s/submission-guidelines#loc-laboratory-protocols. Additionally, PLOS ONE offers an option for publishing peer-reviewed Lab Protocol articles, which describe protocols hosted on protocols.io. Read more information on sharing protocols at https://plos.org/protocols?utm_medium=editorial-email&utm_source=authorletters&utm_campaign=protocols.

We look forward to receiving your revised manuscript.

Kind regards,

Wajid Zaman

Academic Editor

PLOS ONE

Journal Requirements:

Reviewers' comments:

Reviewer's Responses to Questions

**Comments to the Author**

1. If the authors have adequately addressed your comments raised in a previous round of review and you feel that this manuscript is now acceptable for publication, you may indicate that here to bypass the “Comments to the Author” section, enter your conflict of interest statement in the “Confidential to Editor” section, and submit your "Accept" recommendation.

Reviewer #1: (No Response)

Reviewer #2: All comments have been addressed

2. Is the manuscript technically sound, and do the data support the conclusions?

Reviewer #1: Yes

Reviewer #2: Yes

3. Has the statistical analysis been performed appropriately and rigorously? 

Reviewer #1: Yes

Reviewer #2: Yes

4. Have the authors made all data underlying the findings in their manuscript fully available?

Reviewer #1: Yes

Reviewer #2: Yes

5. Is the manuscript presented in an intelligible fashion and written in standard English?

Reviewer #1: No

Reviewer #2: Yes

6. Review Comments to the Author

Reviewer #1: Line 48: Space problem.

55: C. tachangensis F. C. Zhang .The authority name should not italics.

74: Space problem.

78: Same as above.

82: Same as above.

85: Same as above. There must be space between the term “breeding” and citation “[16]”.

Please correct this problem all over the manuscript.

102: No need to mention this subheading “Materials” here.

55-56: Why is this sentence italics?

177: Multiple reaction monitoring (MRM). Capitalize the first letter of the last words.

Table 1: Keep the table in the center or on the left side.

274: among which? Or among them?

335: Why the authors provide citations in the results section?

357: Same as above.

Keep the space between the two lines in the caption of the figure as per journal guidelines.

295: No need to add a full stop in the subheading. Follow journal guidelines.

339: Some text is bold…

389: “Qunfeng Zhang and colleagues (2018)[41]? No need to mention the year. It should be Qunfeng Zhang and colleagues [41].

In the discussion section part please keep the space between the last word of the sentence and the citation. Previously I mention this problem, but the authors ignore…

Reviewer #2: The authors address my concernings already. Anyyway, both Camellia tachangensis and C. sinensis belong to Sect. Thea (L.) Dyer, so they both are coverntaional tea, the former is wild and tea the later is cultivated tea. The usage of genus name Camellia should be more careful, the second on in the text should be C..

7. PLOS authors have the option to publish the peer review history of their article (what does this mean?). If published, this will include your full peer review and any attached files.

Reviewer #1: No

Reviewer #2: No

---

## [Author Response · Author response to Decision Letter 1]

7 Nov 2024

Dear Reviewer,

We thank you for the opportunity to revise the manuscript “Metabolome and Transcriptomics Analyses Reveal Quality Differences between Camellia tachangensis F. C. Zhang and C. sinensis (L.) O. Kunzte” for publication in Plos one. We also thank the reviewers and you for the helpful comments. Following those suggestions, we have made several revisions to the manuscript as outlined below. We look forward to your response and hope the revisions will enable you to accept this manuscript version. In this revised version, changes to our manuscript were all highlighted within the document using red-colored text.

We also appreciate your clear and detailed feedback and hope the explanation fully addresses your concerns. In the remainder of this letter, we discuss your comments individually along with our corresponding responses.

To facilitate this discussion, we first retype your comments in italic font and then present our responses to the comments.

Responses Reviewer #1

Comment 1: Line 48: Space problem

Response 1: Thank you for pointing out our issue. We have added a space between the last word of the sentence in this section and the citation. You can see our modification in line 49.

Comment 2: 55: C. tachangensis F. C. Zhang. The authority name should not italics.

Response 2: Thank you for pointing out our issue. We have made modifications to this section, and the authoritative names are no longer italicized. You can see the changes in line 56.

Comment 3-6: 74: Space problem.

78: Same as above.

82: Same as above.

85: Same as above. There must be space between the term “breeding” and citation “[16]”.

Response 3-6: Thank you for pointing out our issue. Regarding the issue with citation spacing, we have reviewed the entire article and added a space between the last word of sentences and the citations. You can see the changes in lines 74, 79, 83, and 86. All such issues in the manuscript have been corrected.

Comment 7: 102: No need to mention this subheading “Materials” here.

Response 7: Thank you for bringing up this issue. We have deleted the subheading "Materials." You can see this change in line 102.

Comment 8: 55-56: Why is this sentence italics?

Response 8: Thank you for pointing out this issue. The error occurred due to our oversight, and we have now corrected the formatting of this sentence. You can see the changes in lines 56-57.

Comment 9: 177: Multiple reaction monitoring (MRM). Capitalize the first letter of the last words.

Response 9: Thank you for pointing out our error. We have now corrected this word, and you can see the change in line 176.

Comment 10: Table 1: Keep the table in the center or on the left side.

Response 10: Thank you for pointing out our error. We have corrected the table formatting and centered all the tables. You can see this change in line 202.

Comment 11: 274: among which? Or among them?

Response 11: Thank you for pointing out the issue. We have changed "among them" to "among which" in this section. You can see this in line 281.

Comment 12: 335: Why the authors provide citations in the results section?

Response 12: Thank you for raising this issue. The section on the accumulation of organic acids in the results made assumptions and provided literature support. We have moved this section to the discussion part and removed the references from the results section. You can see this change in lines 391-393 of the document.

Comment 13: 357: Same as above.

Response 13: Thank you for pointing out this issue. The section in the results on the biosynthesis of flavonoids made assumptions and provided literature support. We have removed the references from this part of the results. You can see this change in line 352.

Comment 14: Keep the space between the two lines in the caption of the figure as per journal guidelines.

Response 14: Thank you for pointing out the issue. We have changed the formatting of all the captions to double spacing. You can see this change in lines 215-218, 230, 251-252, 289-291, 311-313, 335-340, and 357-361.

Comment 15: 295: No need to add a full stop in the subheading. Follow journal guidelines.

Response 15: Thank you for pointing out the issue. We have reviewed all the subheadings and removed the periods from them. You can see this change in lines 292-293.

Comment 16: 339: Some text is bold…

Response 16: Thank you for pointing out the issue. Regarding the bold font in the captions, we have changed it to regular font. You can see this change in lines 311, 335, and 357.

Comment 17: 389: “Qunfeng Zhang and colleagues (2018)[41]? No need to mention the year. It should be Qunfeng Zhang and colleagues [41].

Response 17: Thank you for raising these issues. We have made changes to this section and removed the year. You can see this in line 384.

Comment 18: In the discussion section part please keep the space between the last word of the sentence and the citation. Previously I mention this problem, but the authors ignore…

Response 18: Thank you for pointing out the issue. I have reviewed the entire text and added a space between the last word of each sentence and the citation. You can see these changes throughout the document.

Responses Reviewer #2

Comment 19: The authors address my concernings already. Anyyway, both Camellia tachangensis and C. sinensis belong to Sect. Thea (L.) Dyer, so they both are coverntaional tea, the former is wild and tea the later is cultivated tea. The usage of genus name Camellia should be more careful, the second on in the text should be C.. 

Response 19: Thank you for your question. We have checked the usage of genus name to Camellia tachangensis ad C. sinensis in the article. Except for the first occurrence where we used the full names all other instances have been changed to C.. 

Thank you again for your positive comments and valuable suggestions to improve the quality of our manuscript. We want to take this opportunity to thank you for all your time involved and this great opportunity for us to improve the manuscript. We hope you will find this revised version satisfactory.

Sincerely,

Lunxian Liu, Xinglin Wang, Qiuying Li, Han Dai, Qihang Zhou, Xingpan Meng, Zhongting Chen, Ximin Zhang, Zhengdong Zhang, Yingliang Liu, Tie Shen and Yin Yi

---

## [Editor Report · Decision Letter 2]

13 Nov 2024

Metabolome and Transcriptomics Analyses Reveal Quality Differences between Camellia tachangensis F. C. Zhang and C. sinensis (L.) O. Kunzte

PONE-D-24-29681R2

Dear Dr. Shen,

We’re pleased to inform you that your manuscript has been judged scientifically suitable for publication and will be formally accepted for publication once it meets all outstanding technical requirements.

Kind regards,

Wajid Zaman

Academic Editor

PLOS ONE
---

## [Editor Report · Acceptance letter]

27 Nov 2024

PONE-D-24-29681R2 

PLOS ONE

Dear Dr. Shen, 

I'm pleased to inform you that your manuscript has been deemed suitable for publication in PLOS ONE. Congratulations! Your manuscript is now being handed over to our production team.

Kind regards, 

on behalf of

Dr. Wajid Zaman 

Academic Editor

PLOS ONE